# Reorganization of Metabolism during Cardiomyogenesis Implies Time-Specific Signaling Pathway Regulation

**DOI:** 10.3390/ijms22031330

**Published:** 2021-01-29

**Authors:** María Julia Barisón, Isabela Tiemy Pereira, Anny Waloski Robert, Bruno Dallagiovanna

**Affiliations:** Basic Stem Cell Biology Laboratory, Instituto Carlos Chagas-FIOCRUZ-PR, Rua Professor Algacyr Munhoz Mader, 3775, Curitiba, PR 81350-010, Brazil; mjbarison@gmail.com (M.J.B.); isaabelaa@gmail.com (I.T.P.); anny.robert@fiocruz.br (A.W.R.)

**Keywords:** hESC, cardiomyocytes, cardiac differentiation, metabolism, RXR heterodimers, thyroid hormone, glycosaminoglycan, lipid homeostasis

## Abstract

Understanding the cell differentiation process involves the characterization of signaling and regulatory pathways. The coordinated action involved in multilevel regulation determines the commitment of stem cells and their differentiation into a specific cell lineage. Cellular metabolism plays a relevant role in modulating the expression of genes, which act as sensors of the extra-and intracellular environment. In this work, we analyzed mRNAs associated with polysomes by focusing on the expression profile of metabolism-related genes during the cardiac differentiation of human embryonic stem cells (hESCs). We compared different time points during cardiac differentiation (pluripotency, embryoid body aggregation, cardiac mesoderm, cardiac progenitor and cardiomyocyte) and showed the immature cell profile of energy metabolism. Highly regulated canonical pathways are thoroughly discussed, such as those involved in metabolic signaling and lipid homeostasis. We reveal the critical relevance of retinoic X receptor (RXR) heterodimers in upstream retinoic acid metabolism and their relationship with thyroid hormone signaling. Additionally, we highlight the importance of lipid homeostasis and extracellular matrix component biosynthesis during cardiomyogenesis, providing new insights into how hESCs reorganize their metabolism during in vitro cardiac differentiation.

## 1. Introduction

During cell differentiation, gene expression and signaling pathways undergo massive changes. It has been shown that metabolic programs are controlled by stage-specific transcription factors and other regulators as well as reciprocal processes of metabolism and commitment to cell fate [1]. Cell fate-influencing mechanisms of cellular metabolism were demonstrated, including how mitochondria, the redox state and metabolic intermediates can affect crucial signal transduction pathways and transcriptional programs [2]. For instance, mitochondrial biogenesis and the associated metabolic shift, which have been described as hallmarks of differentiation processes, are now viewed as active mechanisms affecting stemness and lineage commitment rather than just as consequences of pluripotency and cell differentiation [3].

Cardiomyogenesis responds to complex mechanisms that comprise several regulatory levels, such as transcriptional, translational and epigenetic regulation. Hence, cellular metabolism and metabolic intermediates represent important factors in the regulation of both cardiomyocyte differentiation and maturation [2].

Cardiac differentiation of hESCs is a useful model to understand the molecular and biochemical mechanisms involved in heart development. Importantly, hESC-derived cardiomyocytes (hESC-CMs) represent an essential tool for disease modeling, drug screening and potentially for regenerative medicine [4]. Although hESC-CMs have immature characteristics, resembling those of fetal cardiomyocytes [5], several works have focused on describing differences in metabolic patterns between hESCs and CMs [6,7,8]. However, a complete panorama of the metabolic changes during the developmental stages of in vitro cardiac differentiation is still lacking.

Therefore, the understanding of the processes involved in cardiac differentiation represents the basis for developing optimized protocols for the production of cardiomyocytes in vitro, and the elucidation of the metabolic profile of hESCs undergoing cardiac differentiation could shed light on how specific metabolic pathways could regulate cardiomyogenesis.

To further explore how cellular metabolism participates in cardiomyogenic commitment, we used previously published data from polysome-bound mRNA sequencing performed during in vitro cardiac differentiation [9]. This dataset has already been explored regarding development and cardiac signaling and has been demonstrated to be a powerful tool for searching for genes that are regulated during the cardiac differentiation of hESCs [9,10,11]. Here, we aimed to investigate metabolic changes occurring during the developmental stages of hESC cardiomyogenic differentiation and to elucidate some aspects of how metabolism and its regulation can contribute to cell fate commitment.

## 2. Results and Discussion

### 2.1. Exploration of Cellular Metabolism during Cardiomyogenic Differentiation

The cardiac differentiation process is highly regulated since cells need to exit from the state of pluripotency (hESC), pass through the mesoderm stage and commit to the cardiomyogenic lineage, generating cardiomyocytes. We used polysome-bound RNA-seq data from five stages of in vitro cardiac differentiation (pluripotency (D0), embryoid body (EB) aggregation (D1), cardiac mesoderm (D4), cardiac progenitor (D9) and cardiomyocyte (D15)) (Figure 1A), to investigate the modulation of cellular metabolism during this process. Based on an overall FDR of ≤ 0.05 and −1.5 ≥ logFC ≥ 1.5, we performed comparisons between each differentiation time point and the preceding one (D0 × D1, D1 × D4, D4 × D9 and D9 × D15) to explore the cell transitions (Figure 1A) and identified 707, 2234, 2331 and 1564 differentially expressed genes (DEGs) associated with each transition, respectively. A comparison was also performed between undifferentiated cells vs cardiomyocytes (D0 × D15), which identified 6022 DEGs (Appendix A). As expected, the expression levels of cardiac regulators, structural/sarcomeric genes, calcium-handling genes and ion-channel genes are in agreement with the progression of cardiac differentiation, presenting higher levels at D9 and D15 stages (Appendix A). Then, the DEGs were submitted to analysis using IPA software. The most highly enriched canonical pathways for each comparison were associated with several development and cardiac-related terms, such as “Transcriptional Regulatory Network in Embryonic Stem Cells”, “Regulation of the Epithelial-Mesenchymal Transition Pathway”, “Factors Promoting Cardiogenesis in Vertebrates” and “Calcium Signaling” (Figure 1B and Appendix A). These general analysis corroborate the cardiomyogenic profile and previous results [10,11].

#### 2.1.1. Metabolism-Related Canonical Pathways

To further explore how cellular metabolism participates in cardiomyogenic commitment, we sought to identify canonical pathway-associated terms related to metabolic processes. Figure 2 depicts the metabolism-related pathways associated with each transition. At D0 × D1, we identified the least number of pathways (5) with terms related to RXR signaling pathways (“LXR/RXR Activation” and “LPS/IL-1 mediated inhibition of RXR function”) and amino acid metabolism (“Asparagine Biosynthesis I” and “Glycine betaine degradation”). At the D1 × D4 transition, terms related to lipid metabolism (“Phospholipases” and “Triacylglycerol degradation”), pyrimidines (“Uracil degradation” and “Thymine degradation”), retinol and heparan sulfate biosynthesis and acetone and bupropion degradation were identified (Figure 2).

A higher number of terms related to metabolism was associated with the D4 × D9 and D9 × D15 transitions, where commitment and final differentiation of cardiomyocytes occur. At the D4 × D9 transition, low *p* values were obtained for RXR-related pathways (“FXR/RXR activation”, “LXR/RXR activation”, and other terms), suggesting that metabolism-related signaling plays an important role in the commitment to the cardiomyocyte lineage. Additionally, “retinoate biosynthesis II”, “heparan sulfate biosynthesis (late stages)” and “phospholipases” were associated (Figure 2). At the D9 × D15 transition, RXR-related pathways were also identified, including “TR/RXR activation” and “VDR/RXR activation”, among others, as well as terms related to pyrimidine and lipid metabolism (Figure 2).

When we compared the initial (D0, hESCs) and final points of differentiation (D15, cardiomyocytes), a large number of metabolism-related pathways were identified. Among the more significant terms that were obtained, we again found pathways related to RXR and TR signaling. Additionally, terms associated with metabolic pathways related to the biosynthesis of nucleotides and glycosaminoglycans were present, such as “Pyrimidine ribonucleotides de novo biosynthesis”, “PRPP biosynthesis I”, “Purine nucleotides de novo biosynthesis II”, and “Dermatan sulfate biosynthesis” (Figure 2). Finally, considering the crucial role of lipids in cardiac metabolism, terms related to lipid metabolism were also identified (“Acyl-CoA Hydrolysis”, “Estrogen biosynthesis”, “Palmitate biosynthesis I (animals)” and “Fatty acid biosynthesis initiation II”).

To confirm the metabolism-related pathway enrichment, we performed an additional analysis using the metabolic pathway database BioCyc (Appendix A). This analysis identified pathways similar to those identified by IPA software, such as those related to thyroid hormone metabolism and biosynthesis of retinol, heparan sulfate and lipids. Altogether, these results indicate that we have a valuable dataset that allows us to investigate many aspects of cardiomyogenic differentiation metabolism. We will further explore and discuss the metabolic pathways that are highly regulated during cardiomyogenesis.

#### 2.1.2. Energy Metabolism Changes between hESCs and hESC-CM

Fetal cardiomyocytes depend on glycolysis for energy production during development; however, after birth, the increased energy demand requires a dramatic metabolic switch that reorganizes energy metabolism to enhance mitochondrial oxidative activity based on fatty acid consumption [6]. Although hESC-CMs express specific cardiac markers and show differentiated phenotypes, they have immature characteristics, including those involved in cellular energy metabolism [5]. It is interesting to note that IPA canonical pathways specifically related to energy metabolism were not identified in our analysis (Figure 2). However, considering the substantial metabolic switch required for cellular differentiation, we analyzed the expression levels of genes involved in energy metabolism. Many glycolysis genes are highly expressed during cardiomyogenesis, suggesting that hESC-CMs still depend on glycolysis for energy production (Appendix A). Interestingly, some glycolysis genes present a significant upregulation at the cardiomyocyte stage (D15) when compared to D0, such as glucokinase (GCK), phosphofructokinase P (PFKP), aldolase B (ALDOB) and enolase 3 (ENO3). On the other hand, most of the genes encoding TCA cycle enzymes did not show differential expression during differentiation and only showed a moderate increase (Appendix A). Genes related to fatty acid β-oxidation were increased in expression at D15 when compared to their expression at D0, such as genes encoding long chain fatty acid-CoA ligases 1 and 5 (ACSL1 and ACSL5) and long chain-specific acyl-CoA dehydrogenase (ACADL) (Appendix A). Similar to what was observed for genes encoding TCA cycle enzymes, only a few genes encoding complexes associated with the electron transport chain were significantly upregulated at D15 in comparison to D0 (Appendix A). These results are in line with previous reports regarding the energy metabolism of hESC-CMs. It is well established that hESC-CMs largely depend on glycolysis for ATP production, even under aerobic conditions [12,13], supporting the results showing high expression of glycolytic enzymes observed here. Although hESC-CMs show increased mitochondrial activity and increased expression of TCA and β-oxidation enzymes when compared to undifferentiated cells, they are still metabolically immature when compared to primary fetal heart cardiomyocytes [7,14,15]. Accordingly, the hESC-CMs in our study showed increased expression of TCA, β-oxidation and electron transport chain genes at D15 compared to that at D0, although with only moderate increments. One of the parameters that determines energy metabolism in the cell is substrate availability. hESC-CMs were differentiated in a glucose-rich medium (containing approximately 25 mM glucose), which could contribute to the preference for glycolysis as an energy source. It was shown that high glucose concentrations can reduce mitochondrial function [16], while excluding glucose from the medium favors the metabolic switch to mitochondrial oxidative phosphorylation [13,17]. Also, Feyen and collaborators demonstrated that the inclusion of oxidative substrates to the culture medium of iPSCs-CM produce metabolically mature CMs, improving electrophysiological and mechanical parameters [18].

The data presented here suggest that cardiomyocytes at D15 present a metabolic expression profile compatible with the differentiation conditions and the immature phenotype that already has been described, representing the initial metabolic shift expected in in vitro conditions.

### 2.2. RXR-Dependent Signaling Pathways Are Regulated during Cardiomyogenesis

Interestingly, several signaling pathways dependent on retinoic X receptor (RXR) were associated with the D4 × D9, D9 × D15 and D0 × D15 transitions (Figure 2). RXRs are nuclear receptors that participate in many biological processes, including the regulation of cellular metabolism [19]. They function through their association with several partner receptors by forming heterodimers and integrating with different signaling pathways. Among the three RXR subtypes (α, β and γ), RXRA (α) expression was upregulated in D0 × D15 (logFC 2.32), and RXRG (γ) expression was highly altered during cardiomyogenesis (D1xD4, logFC 7.61; D4 × D9, logFC -3.20, and D9 × D15, logFC 4.98) (Appendix A). It is relevant to note that the regulation of these pathways can occur either via the differential expression of RXR receptors and their partners and/or the expression of their ligands. These partnerships can be classified as (1) permissive when the heterodimers are activated in response to RXR and/or partner ligands, such as peroxisome proliferator-activated receptors (PPARs), farnesoid X receptors (FXRs) and liver X receptors (LXRs), among others; or (2) nonpermissive when the heterodimers are activated by their own agonists and not by RXR agonists, such as retinoic acid receptor (RAR), thyroid hormone receptor (TR) and vitamin D receptor (VDR). Permissive receptors generally function as metabolic sensors for the cell by binding different metabolites with variable affinities [20,21]. Therefore, it is relevant to mention that both metabolic and endocrine mechanisms of regulation are being employed by the cell during cardiac differentiation (Figure 2).

#### 2.2.1. Heterodimers of RXR with PPAR, LXR and FXR Can Modulate Lipid Homeostasis

Among the group of RXR heterodimers highlighted in our analysis (Figure 2), three of them, PPAR, LXR, and FXR, respond not only to hormones but also to molecules related to metabolic pathways. PPARs are nuclear receptors activated by lipids, such as unsaturated fatty acids or eicosanoids, and can regulate different processes of lipid metabolism. Specifically, PPARα is mainly expressed in tissues with high rates of lipid degradation, such as adipose tissue and heart muscle. Accordingly, genes targeted by PPARα comprise those involved in different steps of lipid catabolism, such as fatty acid uptake, β-oxidation and lipoprotein metabolism (reviewed in [22]). LXR is a transcription factor that acts as a sensor of cholesterol levels by targeting genes related to lipid transporters (e.g., ABC transporter family members), apolipoproteins, and genes related to lipogenesis (reviewed by [22,23]). Finally, FXR responds to bile acid levels and regulates bile acid production by regulating the ABC transporter bile salt export pump (BSEP) and other molecules [22,24]. In addition, RXR could also bind and respond to unsaturated natural fatty acids in addition to its traditional ligand 9-cis-retinoic acid [25].

The role of PPARα-RXR in regulating the expression of genes related to lipid and energy metabolism in cardiac tissue is well reported in the literature in both cardiac physiology and in pathological conditions (reviewed in [26,27]). On the other hand, the influence of LXR and FXR on cardiac signaling, metabolism and development is not well understood, although studies have related these pathways to cardiovascular pathologies [23,28,29,30,31].

Comparisons between the PPAR, LXR, and FXR pathways showed that they share many common genes, such as genes encoding apolipoproteins (APOA1, APOA2, APOA4 and APOC1), LDL (lipoprotein lipase), RXRA (α) and FASN (fatty acid synthase). Interestingly, the identified targets are related to lipid catabolism, regulation of cholesterol efflux and transport, and transcriptional activation, among other functions (Appendix A). For instance, MEF2C, which is found in the PPAR/RXR activation pathway, is an essential transcriptional factor involved in cardiac differentiation but can also activate (via the MEF2/HDAC regulatory pathway) the transcriptional coactivator peroxisome proliferator-activated receptor (PPAR)-γ coactivator 1 α (PGC-1α) and consequently affect mitochondrial biogenesis in cardiac tissue [32].

Another RXR heterodimer identified in the canonical pathway analysis was VDR/RXR (Appendix A), which interacts specifically with 1,25(OH)_2_D_3_. The main function of vitamin D is to regulate calcium homeostasis, but other roles have also been described, such as the regulation of proliferation and differentiation [33]. Among VDR pathway targets, several were identified, such as IGFBP1, -5 and -6, RXRG, TGFB2, and other targets (Appendix A). VDR signaling was described as active in cardiomyocytes, where it could play a role in cardiovascular function [34,35]. For example, Wang and colleagues (2001) showed that VDR/RXR signaling is involved in the atrial chamber-specific expression of the low myosin heavy chain 3 gene (slow MyHC3) in heart development [36].

Since at least one of these canonical pathways was highlighted in the IPA results from each time point and all of them appeared in the results for D0 × D15 (Figure 2), we plotted heatmaps of the log2 RPKM values of all DEGs that appeared in the PPARα-, LXR-, FXR-and VDR/RXR activation pathways (Figure 3 and Appendix A). It is interesting to note that, except for the PPARA gene, the FXR, LXR and VDR genes were not identified as differentially expressed in our data (Appendix A), but many of their related pathway genes were shown to be regulated (Figure 3, Appendix A).

Altogether, these results indicate how signaling pathways related to nuclear receptors (RXR heterodimers) could act on different components of metabolism during differentiation, especially on lipids (cholesterol, bile acids, triglycerides, etc.). The promiscuity of RXR activation by several unsaturated metabolites and its association with other lipid metabolite-responding nuclear receptors represent the potential of RXR heterodimers as intracellular sensors of the cell metabolic status [25]. In addition, it is well known that the heart demands a very high level of energy generation to sustain contractile function, ionic homeostasis and basal metabolic processes and obtains most energy from fatty acid oxidation [37]. Our results exemplify that, despite the metabolically immature nature of hESC-CMs, they are subjected to regulated restructuring throughout cardiac commitment, allowing cells to adapt to specific lipid homeostasis.

#### 2.2.2. Stage-Specific Regulation of Retinoic Acid Metabolism

Retinol, or vitamin A, is obtained from the diet and is metabolized into retinoic acid (RA), which has been described as a relevant regulator of gene expression during embryogenesis and specifically during cardiomyogenesis [38,39]. RA activates the RAR nuclear receptors (α, β, and γ), which also form heterodimers with RXRs. Interestingly, canonical pathways related to retinoic acid biosynthesis and signaling were found to be regulated at the mesoderm (D1 × D4) and cardiac progenitor (D4 × D9) stages that were associated with the terms “retinol biosynthesis”, “retinoate biosynthesis” and “RAR activation”. In addition, the same terms were also identified in the D0 × D15 comparison (Figure 2 and Appendix A). Thus, considering the importance of retinoic acid as a morphogen, we describe here the overview of RA metabolism based on our polysome-bound RNA-seq data. The expression pattern of some RA-related genes was validated by qPCR (Appendix A).

Retinol can enter the cell via the STRA6 transporter or by membrane diffusion. Plasma retinol binding protein 4 (RBP4) is responsible for retinol delivery to STRA6, which transfers retinol to the cellular retinol binding protein RBP1 in the cytoplasm [40]. During in vitro cardiomyogenic differentiation, RBP1 was upregulated at D4, RPB4 at D9 and STRA6 at D15 (D0 × D15) (Figure 4). STRA6 functions do not seem to be required for embryo development, when membrane diffusion is the predominant retinol input method. However, retinol transport through STRA6 is essential in adult tissues, such as the heart [40].

Once inside the cell, retinol is oxidized in two enzymatic steps into RA. First, alcohol dehydrogenases (ADHs) or retinol dehydrogenases (RDH) convert, in a reversible reaction, retinol to retinal, which is the aldehyde form. Then, aldehyde dehydrogenases (ALDHs) or retinaldehyde dehydrogenases (RALDHs) oxidize retinal into RA. The enzymes more extensively associated with these functions during embryogenesis and specifically during cardiomyogenesis are RDH10 [41] and ALDH1A2 [42,43], respectively. Genes encoding RDH10 and ALDH1A2 presented higher levels of expression at the mesoderm stage, suggesting an increase in RA synthesis (Figure 4 and Appendix A). At D9, these genes are not differentially expressed, but instead, two other enzymes seem to be downregulated: SDR16C5 (epidermal retinol dehydrogenase 2) and ALDH1A3 (Figure 4 and Appendix A). RDH10 performs most of the embryonic oxidation of retinol, since the Rdh10 null mouse phenotype is very severe. However, there is evidence that points towards the existence of additional retinol dehydrogenases that contribute to RA biosynthesis. It has been shown that the SDR16C5 ortholog in *Xenopus laevis* is essential for frog embryonic development, and murine SDR16C5 is responsible for retinol dehydrogenase activity in skin but is not critical for survival. Human SDR16C5 has low activity but may play a role in adjusting RA cellular levels [44,45,46]. ALDH1A3 is expressed and active in developing embryos; however, it is required for the morphogenesis of facial structures, such as the eyes and nose [47], which could explain its downregulation throughout cardiac differentiation (Figure 4B).

Other factors regulating RA signaling are the cytochrome P450 CYP26A1-C1 enzymes. These enzymes catalyze RA into oxidized retinoids to control RA availability and the extent of RA signaling, which is crucial for embryo anterior-posterior patterning and organogenesis [48]. CYP26 enzymes are upregulated in D4 (CYP26B1), and up- (CYP26B1) and downregulated (CYP26A1) in D9 (Figure 4, Appendix A). Loss-of-function phenotypes for the CYP26 enzymes have been described that affect development [48]; however, the role of individual CYP26 enzymes is not yet clear, especially in mammalian cardiac development. CYP26A1 is the predominant enzyme acting during development, and knock-out mice showed cardiac looping defects [49]. Studies in zebrafish showed that cyp26a1 and cyp26c1 act redundantly to pattern cardiovascular progenitors [50]. In chicks, Cyp26B1 was shown to be expressed in the heart and controlled by RA [51], and it was shown to occupy nonoverlapping sites of expression in the embryo when compared to Cyp26A1 and -C1 [52]. RA acts as a morphogen that regulates the expression of target genes in a concentration-dependent manner. The regions in which signaling by RA is active depend on its synthesis by ALDH1A2 and its degradation by CYP26s. Therefore, the identification of the temporal expression pattern of these key enzymes is relevant for understanding the mechanisms underlying in vitro cardiac differentiation.

Inside the cell, RA and retinal bind to cellular binding proteins, which regulate their availability and cell fate [53]. The intracellular RA binding proteins (CRABP1-2) also presented an interesting pattern. CRABP2 is upregulated at D4, while CRABP1 is progressively downregulated during cardiac differentiation (at D9 and D15) (Figure 4 and Appendix A). The role of these proteins does not seem to be essential for cardiac development [54]. However, CRABP2 acts by shuttling RA into the nucleus and delivering it directly to RARs [40], suggesting its contribution to the RA signaling increase at D4. On the other hand, CRABP1 may sequester RA and shuttle it to catabolizing enzymes [40], controlling RA availability, and its downregulation is in line with the overall upregulation of RA signaling (Figure 4).

Another interesting result of our analysis regarded the regulation of the storage of retinol as retinyl ester. Lecithin retinol acyltransferase (LRAT) esterifies retinol, which can be stored in lipid droplets [55]. In the D1 × D4 transition, LRAT is downregulated, which is an expected result considering that retinol is being actively oxidized to RA. On the other hand, LRAT is upregulated at D9, when RA production from retinol is apparently decreased (Figure 4, Appendix A). This suggests that at D9, retinol could be stored as retinyl esters. On the other hand, the generation of free retinol from retinyl ester stores is not yet established [56]. A group of enzymes called retinyl ester hydrolases (REHs) are proposed to be responsible for this process, and some of them showed distinct expression patterns during differentiation, such as CEL, CES1, LPL and LIPG (Appendix A).

RA acts as a signaling molecule that controls the action of RAR/RXR heterodimers. RARs can act as transcription factors by binding to DNA at specific sequence elements (RAREs, RA response elements). When RA is not present, RAR/RXR bound to RAREs recruits repressive factors that inhibit the transcription of target genes. When RA is synthetized by cells, an activator complex is recruited, and transcription is activated (reviewed by [39]). We found that during in vitro cardiomyogenesis, the isoforms RARβ and RXRγ are differentially expressed at D4 and at D9 (Figure 4, Appendix A). Although it was demonstrated that these receptors present some redundancy in their functions, RXRα null mutants exhibit phenotypes with cardiac defects and embryonic lethality. Additionally, the heterodimer RARα/RXRα is involved in the differentiation of second heart field cells and in the formation of the outflow tract (reviewed in [57]). As previously mentioned, RXRG showed high regulation of expression during cardiomyogenesis, with an expressive difference between D0 and D15 (Figure 4B). To further validate our RNA-seq results, RXRG transcription was evaluated by qPCR using two distinct cardiomyocyte differentiation models: embryoid body and monolayer (Appendix A). As expected, expression level of RXRG was much higher at the end of differentiation (D15) when comparing to D0, in both differentiation models. We also corroborated this result using immunofluorescence assay, which detected RXRγ protein expression at D15, but not at D0 (Appendix A). Although RXRγ could act through its many partners and ligands as we have already discussed above, its direct association with cardiomyogenic differentiation is not clearly understood, and further investigation will be necessary to describe its role in cardiomyogenesis.

The levels of embryonic RA need to be tightly regulated during vertebrate development, as excessive or decreased levels result in many congenital malformations, including heart defects. The establishment of cardiac progenitors in particular, which commit to their fate early in cardiomyogenesis, requires proper RA signaling [57]. Altogether, our data suggest that RA biosynthesis and signaling would be more active at the mesoderm stage (D4), with a slight decrease in cardiac progenitor cells, corroborating the early role of RA in cardiogenic commitment. Interestingly, the main RA signaling activation genes, such as ALDH1A2, CRABP2, RARs and RXRs, were also upregulated in D0xD15, while RA degradation/storage genes, such as CYP26A1, LRAT and CRAPB1, were downregulated in cardiomyocytes, suggesting the overall upregulation of RA signaling throughout cardiac differentiation (Figure 4B).

#### 2.2.3. TH Metabolism and Signaling Influence Downstream Pathways Regulating Cardiomyogenesis

Thyroid hormones (THs) play important roles in development and heart function. Altered levels of maternal TH during gestation can cause abnormal embryo development, affecting, for example, proper heart formation. TH levels are maintained at low levels during fetal development [58], which are required for proper cardiomyocyte proliferation. Before birth, the T3 concentration increases (prenatal surge) and contributes to cardiovascular and metabolic adjustment once T3 participates in the maturation of cardiomyocytes and the formation of normal heart physiology (reviewed in [59] and [60]).

TH action is initiated through the binding of T3 to thyroid receptor (TR). The IPA identified the canonical pathway “TR/RXR activation” in the D9 × D15 and D0 × D15 transitions (Figure 2 and Appendix A), although some differentially expressed genes related to TH metabolism and signaling were also identified in D4 × D9 (Appendix A). The expression pattern of some TH metabolism genes was also corroborated by qPCR (Appendix A).

Inside the cell, TH can be metabolized by specific enzymes called deiodinases, which can produce active (T3) or inactive (rT3, T2) forms of TH (Figure 5). DIO2 produces the active form of TH, T3, through the deiodination of the prohormone T4. On the other hand, DIO3 inactivates T3, generating T2, or deiodinates T4, producing rT3, avoiding its activation. DIO1 has a dual function in T4 metabolism; it activates (T4 to T3) or inactivates the prohormone (T4 to rT3), though with reduced kinetic efficiency. Therefore, deiodinase activity is a relevant regulator of T3 availability inside the cell by modulating the pool of active or inactive TH [61]. DIO2 expression increases during cardiomyogenesis and shows a peak at D9 (D4 × D9, logFC 3.808, Appendix A) (Figure 5B,C and Appendix A), suggesting that prohormone activation (T4 to T3) may occur at the cardiac progenitor stage. DIO2 is expressed in the human heart, and its expression is regulated by the master cardiac transcription factors Nkx-2.5 and GATA-4 [62]. Interestingly, DIO2 is downregulated in cardiomyocytes compared to cardiac progenitors (Figure 5B and Appendix A). Additionally, DIO3 is upregulated on the first day of differentiation (D0 × D1, logFC 3.131, Appendix A), and its expression is maintained during cardiomyogenesis (Figure 5B and Appendix A). It has been shown that temporal and tissue-specific expression of DIO2 and DIO3 is crucial for cellular proliferation and differentiation during development [63]. However, whether coordination of DIO2 and DIO3 activities contributes to T3 homeostasis in the heart is not clear. The expression of DIO3, which restrains TH signaling, substantially decreases after birth in cardiac tissue [64]. This decrease coincides with the T3 prenatal surge, indicating the crucial role of DIO3 in controlling TH signaling during development. Chattergoon and colleagues, in a collection of works, suggested that there is a narrow fetal TH concentration range required to produce a normal heart that ensures the balance between proliferation and maturation of cardiomyocytes [60]. Our data show that DIO3 is highly expressed during in vitro cardiomyogenesis (Figure 5B and Appendix A), which could be critical to control TH signaling during the early steps of differentiation.

Nevertheless, TH signaling is crucial for cardiomyocyte maturation and heart function. Using murine ESCs, Lee and collaborators showed that T3 improved in vitro cardiomyogenesis while producing a more adult CM phenotype [65]. In humans, Yang and colleagues also showed that T3 treatment favors hiPSC-CM maturation by improving, among other parameters, the maximal mitochondrial respiratory capacity, suggesting a more mature metabolic profile [66]. TH actions depend on the interaction with TR and RXR. The resulting heterodimer regulates gene expression by binding to specific sequences in the genome denoted as TREs (thyroid hormone response elements) [67]. In addition to RXR (Figure 4B), thyroid hormone receptor alpha (THRA) was also found to be upregulated in cardiomyocytes (Figure 5 and Appendix A). This is in accordance with the fact that among the two major isoforms of TR (THRα and THRβ), the alpha isoform is responsible for most of the effects on cardiac genes [68]. In addition to THRα and THRβ, it was shown that a truncated isoform of THRα (receptor p43) plays a relevant role in mitochondrial metabolism. Casas and colleagues demonstrated that in mouse skeletal muscle, the receptor p43, which resides in the mitochondrial matrix and responds to T3, activates mitochondrial genes, increasing respiratory rates and inducing a metabolic switch favoring increased oxidative metabolism [69,70]. Whether this receptor plays a role in metabolic restructuring in cardiomyocytes and mitochondrial function has not yet been described.

In accordance with the upregulation of its receptors, the TR/RXR activation pathway was associated with a higher number of DEGs in the D0 × D15 transition analysis. All three deiodinases were upregulated in the CM stage. Interestingly, monocarboxylate transporter 8 (MCT8, SLC16A2), which is the only transport system described as specific for THs [71], was decreased in expression between D0 and D15 (Figure 5C and Appendix A).

Genes related to fatty acid biosynthesis, lipolysis, and cholesterol, carbohydrate and steroid metabolism were upregulated between D0 and D15 (Figure 5A,C, Appendix A), suggesting that signaling through TR/RXR could play a relevant role in restructuring metabolism during cardiomyogenesis. For instance, BCL3, APOA5 AKR1C1-2 and PCK1 were also upregulated in D9 × D15 (Figure 5C and Appendix A), indicating that TH signaling could act in late stages of differentiation, specifically in the progenitor-cardiomyocyte remodeling of metabolism. Among the genes regulated by TH that are related to carbohydrate metabolism, we found PFKP (ATP-dependent 6-phosphofructokinase, platelet type isoform), SLC2A1 (glucose transporter GLUT1) and SLC16A3 (monocarboxylate transporter 4, MCT4), which are all related to glucose utilization. PFK is a regulatory glycolytic enzyme catalyzing the phosphorylation ATP-dependent of fructose-6-bisphosphate and producing fructose-1,6-bisphosphate. Interestingly, the PFKP isoform that was the only one found to be differentially expressed based on our data (Figure 5) also plays roles in processes other than glycolysis, such as the modulation of MAP kinases [72]. GLUT1 and MCT4 are membrane transporters that are also involved in glucose metabolism. André and colleagues demonstrated that both transporters are expressed in cardiac progenitors and mature neonatal rat cardiomyocytes, while glucose consumption and lactate release were increased in CPs [73]. The upregulation of GLUT1 and MCT4 observed in our CM data could be another indication that hESC-CMs present a higher dependence on glycolysis for energy production as a consequence of their immature phenotype.

Cytosolic PCK1 is one of the TR/RXR target genes upregulated in the D9 × D15 and D0 × D15 transitions (Figure 5 and Appendix A). This enzyme is a well-known gluconeogenesis regulator and a key enzyme in glyceroneogenesis. PCK1 presents cataplerotic and anaplerotic activities and is expressed mainly in the liver, adipose tissue and kidney but is also present in other tissues [74]. To the best of our knowledge, PCK1 expression in cardiomyocytes has not yet been shown. Gluconeogenesis is not a major process in cardiac muscle, and the metabolic functions and regulatory mechanisms of PCK1 beyond those involved in gluconeogenesis are poorly understood. An interesting study involving the overexpression of PCK1 in skeletal muscle showed that transgenic mice had an increased oxidative capacity with a significant increase in mitochondrial biogenesis and triglyceride content [75]. The remodeling of metabolism induced by overexpression of PCK1 suggests that this enzyme could be involved in restructuring the cell bioenergetics in response to changes in the cellular microenvironment and energy demands. Based on these data, PCK1 has become an interesting target of study to explore whether its increased expression in cardiomyocytes (Figure 5 and Appendix A) is related to the specific roles of this enzyme in cataplerotic/anaplerotic reactions, glyceroneogenesis and/or energetic metabolism.

Here, we show that genes related to the TR/RXR pathway are differentially regulated from the mesoderm stage to the CM stage and suggest that through genomic action, T3 could activate the expression of several targets (e.g., PCK1), which can have some influence on the metabolic switch observed during cardiac differentiation. In addition, the regulated expression of deiodinases could play a crucial role in controlling TH signaling and ensuring the balance of proliferation and maturation of cardiomyocytes during development.

### 2.3. Additional Lipid Metabolism-Related Pathways Are Enriched during Cardiac Differentiation

We have already discussed how diverse signaling pathways related to RXR heterodimers could act in lipid metabolism during differentiation. Another canonical pathway involved in lipid metabolism found in our analysis was associated with the term “phospholipases” (Figure 2), which represent an important group of enzymes involved in fatty acid metabolism.

Phospholipases hydrolyze fatty acids from phospholipids, and the enzyme subtype depends on the catalyzed reaction. A total of 10 enzymes were identified in D1 × D4: 6 upregulated and 4 downregulated enzymes. Additionally, at the D4 × D9 transition, 11 enzymes were differentially expressed, with 6 upregulated and 5 downregulated enzymes (Figure 6A).

Upon comparing the DEGs between both groups, 4 enzymes were shown to be regulated in the D1 × D4 and D4 × D9 transitions in opposing ways (Figure 6A), which could suggest their specific role during cardiac differentiation. Among them, we found NOTUM and PLA2G2A, which had increased expression at the mesoderm stage and decreased expression at the CP stage (Figure 6A). NOTUM was described as an extracellular phospholipase involved in deacylation of Wnt proteins and suppression of Wnt signaling [76]. Wnt signaling has been shown to play multiple and opposing roles during cardiac differentiation and development, especially in mesoderm and cardiac specification [77]. Therefore, NOTUM may participate in the strong and expected regulation of Wnt signaling in our model of cardiomyogenic differentiation. PLA2G2A is an enzyme that belongs to the secretory phospholipase A2 family and catalyzes the hydrolysis of the acyl groups in the sn-2 position of phospholipids, producing fatty acids and lysophospholipids [78,79]. Several functions have been described for this enzyme, such as binding and activation of integrins [80], synthesis of phosphatidic acid [81] and pro-inflammatory action in atherosclerosis [82]. Interestingly, it was shown that this secreted phospholipase regulates the expression of proteins related to energy metabolism in adipose tissue [83].

PLA2G4C and PLB1 (phospholipase B1) are phospholipases associated with membranes and were also regulated in the D1 × D4 and D4 × D9 transitions (Figure 6A). PLA2G4C (cPLA2-gamma) was first identified in [85] and [86]. It also presents PLA1 [86] and lysophospholipase activity [87,88]. Furthermore, Yamashita and colleagues demonstrated that this enzyme presents transacylation activity and could be found in the endoplasmic reticulum and mitochondria [89]. Additionally, it was shown that this enzyme is mainly expressed in skeletal muscle and heart [86], corroborating our data (Figure 6A). Interestingly, we observed that this gene is differentially expressed at all transitions evaluated, not only at D1 × D4 and D4 × D9, suggesting that PLA2G4C could exert relevant functions during differentiation and is subject to tight regulation.

Considering the different aspects of lipid metabolism described here for phospholipases, such as PLA2G2A and PLA2G4C, and the fact that they are regulated throughout the cardiomyogenesis process, we propose that these enzymes could be considered interesting candidates for further investigation.

Moreover, phospholipases are also involved in the metabolism of phosphatidic acids (PAs), which are specific phospholipids in which a phosphate group occupies the C3 position of the glycerol molecule. PAs can act as precursors for glycerophospholipid and triacylglycerol biosynthesis. In this case, PA is generated from de novo synthesis through two acylation reactions from glyceraldehyde-3-phosphate (G3P) (Figure 6B). Additionally, PAs play an important role as secondary messengers in several signaling pathways. In this case, PA can be produced by different pathways: (i) production of diacylglycerol (DAG) by phospholipase C (PLCs) and subsequent phosphorylation of DAG by DAG kinases (DGKs), (ii) by the action of phospholipase D, which hydrolyzes phospholipids, mainly phosphatidylcholine, to produce PA or (iii) by lyso-PA acyltransferases [84,90]. In addition, the dephosphorylation of PAs to generate DAGs is carried out by phosphatidic acid phosphatases (PAPs) (Figure 6B) [91]. Nevertheless, the action of DGKs and PAPs controls lipid homeostasis, regulating the balance between PA and DAG.

Therefore, we performed a detailed search for DEGs encoding enzymes related to the phosphatidic acid metabolism pathway during cardiomyogenesis and observed interesting gene expression profiles for many genes (Figure 6C and Appendix A).

As previously mentioned, PA can be synthetized from glycerophospholipids through the action of phospholipases C (PLCs) and DAG kinases. We found 7 differentially expressed PLCs with specific expression patterns (Figure 6C and Appendix A). An interesting case is PLCXD3 (phospholipase C X-domain containing protein), which presents increased levels at the CM stage (Figure 6C). Phospholipase C is mainly involved in the production of the secondary messengers DAG and inositol 1,4,5-trisphosphate (IP_3_), therefore participating in several signaling pathways [92]. Specifically, PLCXD3 presents only the catalytic X domain and distinct intracellular localization in the cytoplasm and perinuclear vesicles, suggesting that it could be involved in different functions [93]. Additionally, the authors demonstrated that PLCXD3 is expressed mainly in the brain and heart, corroborating our results.

Different isoforms of DGKs showed specific expression profiles during differentiation (Figure 6C and Appendix A). For instance, DGKι is a very interesting subtype, as it presented higher levels at the cardiac progenitor stage and decreased levels in cardiomyocytes. DGKι was described as a cytosolic and nuclear enzyme that is present mainly in the nervous system [94,95], but nothing has been reported about its expression and function in cardiomyocytes. Interestingly, some DGKs have been shown to be expressed in rat heart (DGKα, ε and ζ, [96]), but none of them have been identified as DEGs in our dataset.

PAPs are also enzymes related to the PA-DAG balance. In our dataset, we found 3 differentially expressed PAPs: PPAP2C, PPAPDC3 and PPAPDC1A (Figure 6C and Appendix A). Specifically, PPAP2C and PPAPDC1A expression were decreased between D4 and D9, while PPAPDC3 expression was increased at D15 compared to D0 (Figure 6C).

The de novo synthesis of PA from G3P involves the activity of glycerol-3P acyltransferases (GPATs) and 1-acyl-sn-glycerol-3-phosphate acyltransferases (AGPATs) (Figure 6B). The only differentially expressed GPAT gene was GPAT3, presenting an increasing expression pattern between the CP and CM stages (Figure 6C and Appendix A). The identified AGPATs were AGPAT2 and AGPAT3, which presented a significant fold change in expression between D0 and D15 (Figure 6C and Appendix A). Interestingly, glycolytic enzymes involved in the first steps of the pathway leading to G3P production showed some level of upregulation in D0 × D15 (HK2, logFC 1.36; GCK, logFC 3.68; GPI, logFC 1.41; PFKP, logFC 2.82; ALDOA, logFC 1.68; ALDOB, logFC 5.67). Together, these results suggest that at the CM stage, some PA could be synthetized from G3P. Again, differentiation medium composition seems to be influencing cardiomyocyte metabolism. As previously described, StemPro-34 presents high glucose concentrations (about 25 mM = 4.5 g/L), while low lipid contribution (around 5 mg/L). In this way, glycolytic intermediates as G3P could be used by CMs to direct the synthesis of PA, a precursor for lipid biosynthesis.

### 2.4. Heparan Sulfate Biosynthesis: Modification Enzymes and Cell Fate Regulation

Heparan sulfate (HS), a glycosaminoglycan (GAG), is a sulfated polysaccharide that modifies membrane and extracellular matrix (ECM) proteins. These proteoglycans have important functions in interacting with morphogens and growth factors and therefore in regulating developmental processes. Biosynthesis of HS and other GAGs involves three main steps: glycosaminoglycan-protein linkage synthesis, polymerization and modification (Figure 7) (reviewed in [97]).

Using IPA Canonical Pathway analysis, the terms “Heparan Sulfate Biosynthesis” and “Heparan Sulfate Biosynthesis (late stages)” were obtained for the D1 × D4 and D4 × D9 transitions (Figure 2 and Appendix A), suggesting that genes encoding enzymes involved in these stages are differentially regulated during cardiomyogenesis. Figure 7 represents the heparan sulfate biosynthesis pathway and the genes up- or downregulated at the mesoderm (D1 × D4) and cardiac progenitor (D4 × D9) stages, while Appendix A presents the expression pattern validation of some HS-genes obtained by qPCR at D1, D4 and D9 stages. At D4, it was observed that B3GAT2 (galactosylgalactosylxylosylprotein 3-beta-glucuronosyltransferase 2), the enzyme that catalyzes the last step of glycosaminoglycan-protein linkage synthesis, and exostosin-like 1 (EXTL1), a glycosyltransferase related to HS biosynthesis at the polymerization step, were upregulated, as well as the HS-modifying enzymes (HS3ST3A1 and HS3ST3B1). On the other hand, at D9, B3GAT2 was found to be downregulated as well as two other enzymes involved in the last steps of HS biosynthesis (NDST3 and HS3ST3A1) (Figure 7 and Appendix A).

Several studies have shown how the heparan sulfate composition changes during the differentiation of hESCs. For instance, decreased N-sulfation was found in undifferentiated hESCs, while the expression of modifying enzymes, such as N- or O-sulfotransferases, were increased in the transition to committed specific cell lineages. As a result, there is an increased level of sulfation in HS, which appears to be cell-type specific and influences the cell fate (reviewed in [99]). In addition, the expression of HS3ST (HS 3-O-sulfotransferase) varies during development and in different organs. It was demonstrated that FGF10/FGFR2b regulates the expression of HS3STs in fetal salivary gland KIT+ progenitors and that exogenous 3-O-HS could be capable of expanding fetal and adult KIT+FGFR2b+ progenitors in vitro [100]. Accordingly, our results suggest that at the mesoderm stage, HS could be modified by HS-glucosamine 3-O-sulfotransferases (HS3ST3A1 and HS3ST3B1) but not by HS-glucosamine 3-sulfotransferases (HS3ST2), generating specific modifications that could be related to upstream cell commitment. At the cardiac progenitor stage, some synthesis of HS still occurs, but specific modifications decrease, such as N- or O-sulfation.

Heparan sulfate proteoglycans (HSPGs) are a class of glycoproteins that contain one or more covalently attached HS chains and are classified based on their location as membrane HSPGs, secreted ECM HSPGs or secretory vesicle proteoglycans. These glycoproteins can bind growth factors, morphogens and other molecules, protecting them from degradation and generation of morphogen gradients essential for development, for example (reviewed by [101]).

Perlecan (HSPG2) is a proteoglycan essential in basement membranes that binds with other ECM proteins and growth factors. It was previously shown that the absence of HSPG2 caused embryonic lethality in mice, mainly by causing heart defects, and that this protein is necessary for the stability of heart tissue [102]. The expression of the HSPG2 gene increased during cardiac differentiation (Appendix A), reaching a higher level at D15 and confirming its importance in cardiac commitment. Furthermore, recently, it was demonstrated that perlecan could also regulate metabolism in adipose and skeletal muscle tissues by increasing fatty acid oxidation in adipose tissue and changing the metabolic profile of muscle fibers from glycolytic to oxidative in a perinatal lethality-rescued perlecan knockout mouse model [103].

A decrease in GAG chain sulfation via the agrin signal transduction pathway leads to a diminished frequency of acetylcholine receptor clustering in C2C12 cells [104]. Agrin was also described as capable of promoting proliferation, but it also suppressed the maturation of human iPSC-derived cardiomyocytes [105]. In addition to the regulation of sulfotransferases (Figure 7), the AGRN gene was also modulated during in vitro cardiomyogenesis (Appendix A).

Taken together, these data demonstrate that the correct formation of proteins modified with heparan sulfate chains is important for cardiac differentiation. The modifications performed by enzymes involved in HS synthesis could change the responses of the proteins in the microenvironment, and in this study, we identified some of the proteins and enzymes that are modulated during cardiac differentiation.

### 2.5. Biological Functions Related to Metabolism are Stage-Specifically Regulated

We performed an additional analysis of the DEGs at each transition to identify enriched biological functions using the “Disease and Function” tool in IPA software. We filtered only terms related to cellular metabolism (see Material and Methods). Figure 8 depicts the identified biological functions and the number of up- or downregulated genes for each function.

At the D0 × D1 transition, the terms with the highest number of regulated genes were “Secretion of molecule”, “Synthesis of cyclic AMP” and “Biosynthesis of cyclic nucleotides”, which were associated predominantly with upregulated genes. Additionally, terms related to lipid metabolism, such as “Depletion of phosphatidic acid”, “Secretion of lipid”, and “Cholesterol transport”, were observed. In the mesoderm commitment stage, at D1 × D4, the more greatly enriched biological functions were “Synthesis of carbohydrate” and “Secretion of molecule”, followed by “Metabolism of terpenoid” and “Transport of (monovalent) inorganic cation”. At the D4 × D9 transition, interestingly, only 3 terms were identified, all related to transport/secretion processes: “Transport of molecule”, with almost 200 involved genes, presented predominantly upregulated genes; “Transport of ion” and “Secretion of molecule” appeared to be associated with ~100 and ~75 differentially expressed genes, respectively (Figure 8).

In the cardiac stage, at D9 × D15, the term associated with the highest number of differentially expressed genes was, once again, “Transport of ion”. In this case, most genes were downregulated. In addition, many metabolism-related terms were identified, such as “Metabolism of carbohydrate/polysaccharide” and “Fatty acid metabolism”, which were associated with a large number of DEGs (Figure 8). Last, in the D0 × D15 analysis, the highest number of DEGs was associated with the term “Metabolism of carbohydrate”, which was linked mainly to upregulated genes.

Aiming to identify the most highly differentially regulated processes during cardiomyogenesis, we searched for “Disease and function” terms that were shared between the different comparisons. Among them, “Transport of ion”, carbohydrate-related terms (“Synthesis of polysaccharide”, “Synthesis of carbohydrate”, “Secretion of carbohydrate” and “Metabolism of polysaccharide”), and “Secretion of molecule” were highlighted. Then, the genes shared between different time point analyses were plotted in a heatmap and grouped by hierarchical clustering to better visualize their expression patterns (Figure 9).

The term *“transport of ion”* appeared for the D4 × D9 and D9 × D15 transitions, specifically when the commitment to cardiomyocyte differentiation was initiated. There were 70 specific DEGs involved in the mesoderm-cardiac progenitor transition, while 28 DEGs were specifically associated with the progenitor-cardiomyocyte transition. Interestingly, 17 DEGs were associated with both transitions, suggesting the greater regulation of their expression during cardiac differentiation (Figure 9A). Some genes showed an expression pattern that decreased during differentiation, such as NPPA, TRPC4, GJA1 and SLC4A11, and others showed an increase in their expression, such as ACTN2, AQP1, KCNJ8 and FGF23. Interestingly, a set of genes, including SCN9A, SGK1, KCNMB1, ATP6V0D2, KCNAB1, RAMP2, SLC9A9 and CCR1, showed the maximal expression at D9, while CHRNB4 showed maximal expression at D4.

Effective cardiac function is ensured by the appropriate rate and timing of contraction. The cardiac electrical system is responsible for the specific action potential properties and results from the expression patterns of ion channels. For example, KCNJ8 encodes an inward-rectifier type potassium channel expressed in heart tissue [106], and it is upregulated at D15. On the other hand, KCNMB1 encodes a modulatory subunit of a large conductance calcium-activated potassium channel (MaxiK) and modulates the calcium sensitivity and gating kinetics of the pore-forming subunit. The expression of this gene is restricted in smooth muscle cells, mainly in the vascular cells, in which fine-tuning of vascular tone is performed [107]. KCNMB1 showed a peak in expression in progenitor cells (D9), which are able to differentiate into vascular cells. However, its expression decreased in cardiomyocytes (D15), even though it is not specifically expressed in these cells. Other ion channel genes also appeared to be specifically upregulated at D9, such as SCN9A (voltage-gated sodium channel, a major contributor to pain signaling [108] and SLC9A9 (regulates the luminal pH of the recycling endosome) [109], and their role in cardiomyogenic differentiation has not yet been described.

Some genes related to carbohydrate metabolism also showed a differential expression pattern during cardiomyogenesis. Between D1 and D4, 40 specific genes were differentially expressed, while 43 genes were differentially expressed between D9 and D15. Fifteen genes were regulated in both the D1 × D4 and D9 × D15 transitions, but while some genes were specifically associated with mesoderm stages (e.g., HAS2, APLNR, and CCKBR), others were more highly expressed in cardiomyocytes (e.g., PPI1R3C, IGF2, FMOD and others) (Figure 9B).

Regarding carbohydrate metabolism, glucose homeostasis has a relevant importance. Glycogen serves as a repository of glucose in many tissues, and its function in the heart extends beyond serving as an endogenous fuel. This molecule also has a protective effect on heart function by acting as part of the complex metabolic signaling involved in cell survival and Ca^2+^ homeostasis [110] as well as in normal heart development [111]. We found that the expression of the PPP1R3C gene continuously increased during cardiac differentiation, reaching the highest level at D15. It encodes protein targeting to glycogen (PTG), which is a scaffolding protein that directs protein phosphatase 1 (PP1) to glycogen, reducing glycogen phosphorylase activity and affecting enzymes involved in glycogen synthesis and degradation [112].

The protein phosphorylation and dephosphorylation system, which controls the flux of glucose into and out of intracellular glycogen stores, is regulated by hormones, such as insulin [110,113]. IGF2 encodes insulin-like growth factor 2 (IGF-II) and shows an increase in its expression pattern during cardiac differentiation. Its role in growth and development is very well documented [114], including in heart development and cardiomyocyte proliferation [115]. However, IGF2 functions in cellular metabolism, and the relation of this to its role in development is poorly understood. There have been suggestions about its function as a metabolic regulator (reviewed by [116]) and the role of IGF2 in cartilage development and glucose metabolism [117]. In addition to IGF-II mRNA, the IGF2 locus encodes antisense RNA, long noncoding RNA, and several microRNAs, which could play important and independent roles in metabolic regulation [116].

Among the DEGs, we also found two different hyaluronan (HA) synthase genes, HAS2 and HAS3. HA is a glycosaminoglycan that acts as a structural component of extracellular matrices and promotes a variety of cellular signaling pathways by interacting with cell surface receptors [118]. HAS3 expression has already been described in hESCs and shown to be downregulated in hESC-CMs [119], corroborating our data (Figure 9B). HAS2 was shown to be highly expressed in mesodermal cells of the developing limb bud [118], which is derived from the same progenitors as the cardiac mesoderm [120], and this could explain the increase in expression that we observed at D4 (Figure 9B). Additionally, the importance of Has2 during mouse cardiovascular development has already been shown [121]. Deletion of the Has2 gene resulted in embryonic death and demonstrated the crucial role of HA in the stimulation of cardiac endothelial cell migration.

Interestingly, another gene with high expression in D4 was APLNR (D1 × D4, logFC 7.75). It encodes the APJ/Apelin receptor, a G-protein coupled receptor that is activated by the Apelin peptide. This signaling pathway participates in a wide range of physiological effects, including metabolism regulation, via insulin secretion, glucose uptake and fatty acid oxidation (reviewed by [122]). In addition, the APJ/Apelin receptor plays an important role in vasodilation, angiogenesis and heart development (reviewed by [123]). It has been shown that the zebrafish ortholog is broadly expressed in gastrulating mesendoderm but not in the later stages of myocardium development, suggesting its role in early myocardial progenitor commitment [124], corroborating our data. Notably, the function of Aplnr in the context of cardiac progenitor development occurs independently of the G-protein mechanism and the canonical Apelin ligand [125]. In our data, the Apelin gene (APLN) was not detected as a DEG (data not shown). Therefore, the high expression of APLNR during cardiomyogenesis seems to be more highly related to its role in the signaling related to cardiac progenitor commitment than to metabolism regulation.

A third term that was associated with genes differentially regulated during cardiomyogenesis was “Secretion of molecule”. Interestingly, in this case, we identified genes differentially regulated during differentiation from D0 to D9. Four, nineteen and thirty-one genes were regulated exclusively at the D0 × D1, D1 × D4 and D4 × D9 transitions, respectively. Eight genes were differentially expressed at the three transitions, including ANXA1, DRD2, MYOM1, A2M and other genes (Figure 9C).

The cardiogenic niches are highly dynamic, presenting different functionalities and characteristics according to the stage of development of the heart and its physiological state [126]. For example, ECM components, bioactive fragments released from ECM by enzymatic degradation, growth factors that use ECM as a reservoir and other bioactive molecules released from cells contribute to the function of and signaling by the niche [127]. In the “Secretion of molecule” group of genes, A2M showed continuously increasing expression during cardiomyogenesis. Alpha-2 macroglobulin (A2M) is secreted from cells and has the ability to inhibit proteases, such as collagenases and metalloproteases (MMPs), using a bait and trap method to contribute to niche dynamics. Its capacity to modulate MMP activity in vascular smooth muscle cells and its role in ECM accumulation have been shown [128].

In the same group of DEGs, Annexin A1 (ANXA1), which belongs to the superfamily of calcium-dependent phospholipid-binding proteins, could inhibit the action of cytosolic phospholipase A2 and subsequent inflammatory responses (recently reviewed by [129]). This gene showed a highly regulated expression pattern during cardiomyogenesis (Figure 9C), which could be related to the highly regulated expression of phospholipases, as previously discussed (Figure 6). However, the widespread effects of the ANXA1 protein and its peptide on other cellular processes have been recently recognized [130]. One of these effects is the ability of ANXA1 to interact with integrins and regulate cell adhesion and migration, which could be implicated in cardiac niche signaling. In addition, its peak expression was observed at D1 (D0 × D1, logFC 4.99), which represents embryoid body formation (under hypoxic conditions in our differentiation method) and the exit from the pluripotent state. It has been shown that ANXA1 expression is increased by hypoxic stimuli and plays a crucial role in epithelial to mesenchymal transition (EMT) in cancer cell lines [131]. EMT is also a critical process during differentiation, and the role of ANXA1 should be better explored in the cardiomyogenic context.

## 3. Concluding Remarks

In this study, we analyzed the polysome-bound mRNA-seq data of in vitro cardiac differentiation from a metabolic point of view. The gene expression profile confirmed that hESC-CMs differentiated for 15 days show immature energy metabolism that is highly dependent on glycolysis. Canonical pathway enrichment analysis showed many metabolism-related pathways at each transition in cardiac differentiation. Several signaling pathways involved in the modulation of metabolism were identified, such as the PPARα/RXR and LXR/RXR pathways, suggesting their relevant role in metabolism restructuring, especially in lipid homeostasis. Additional pathways related to lipids, which are key molecules involved in cardiac metabolism, were also analyzed; the differential expression of phospholipases and other enzymes interacting with phosphatidic acid were detected and showed specific expression patterns. The action of the T3 hormone was also described, and we presented several interesting aspects of its metabolism and its contribution to gene expression regulation in cardiomyogenesis. We further analyzed the biosynthesis of molecules acting on differentiation: production of retinoic acid, a potent morphogen involved in development, and the synthesis of glycosaminoglycans such as heparan sulfate. Finally, genes involved in specific biological functions related to metabolism were also described, confirming the regulation of important cellular processes discussed here, such as glucose homeostasis and ECM component synthesis and signaling. Our analysis offers a detailed description of metabolic and signaling pathways relevant to metabolism modulation at specific time points during in vitro cardiomyogenesis. This work provides new insights into how hESCs regulate and reorganize their metabolism during cardiac commitment, allowing the improved optimization of protocols to obtain CMs in vitro.

## 4. Materials and Methods

### 4.1. Cardiomyocyte Differentiation, Polysome Profiling and Sequencing

The NKX2-5eGFP/w HES3 cell lineage was a donation from Monash University (Victoria, Australia) [132]. hESCs were cultured and subjected to a cardiac differentiation protocol described in detail in [9]. Cell lysates from monolayers of hESCs on day 0 (D0) or differentiating EBs (on days 1, 4, 9 and 15, referred to as D1, D4, D9 and D15, respectively) were fractionated using 10% to 50% sucrose gradient in an ISCO gradient fractionation system (ISCO Model 160 Gradient Former Foxy Jr. Fraction Collector, City of Industry, CA, USA), as described in [9]. The polysome-bound fractions (fractions 10–22) were pooled, and the RNA was isolated. For sequencing, the RNA was used to prepare cDNA libraries, and RNA-seq was carried out on an Illumina HiSeq platform. The datasets generated and analyzed during the current study are available in the Short Read Archive, NCBI (https://www.ncbi.nlm.nih.gov/sra/SRP150416).

### 4.2. Bioinformatic RNA-seq Analysis

The reads were mapped against the reference genome GRCh38 with Rsubreads, and the features were counted using the function featureCounts as previously described [9]. Differential expression analysis was performed using the Bioconductor R package edgeR [133]. Comparisons of each sample against the sample from the preceding time-point were performed: D0 vs. D1, D1 vs. D4, D4 vs. D9, and D9 vs. D15; a comparison was also performed between D0 vs. D15. For these analyses, we retained only those genes with at least one count per million in at least three samples. The recommended methods estimateGLMCommonDisp, estimateGLMTrendedDisp and estimateGLMTagwiseDisp were used for normalization. Differential expression analysis was performed using the generalized linear mixed model (glmFit and glmLRT). Correction for multiple testing was performed according to the false discovery rate (FDR), and the RPKM values for each sample were also determined to compare the expression between samples.

### 4.3. Data Analysis

Data analysis was performed using Ingenuity Pathway Analysis (IPA) software (Qiagen Inc. https://www.qiagenbioinformatics.com/products/ingenuity-pathway-analysis). A file with the differentially expressed genes (DEGs) from each comparison (Figure 1A) that included the gene ID, log2(FC) and FDR values, was uploaded. According to the criteria of log2(FC) ≥ 1.5 for upregulated genes and log2(FC) ≤ −1.5 for downregulated genes, the canonical pathways identified from IPA were selected according to their relationship to metabolic processes, and the terms from the Disease and Function analysis were filtered according to the terms “metabolism” and “molecular” (terms related to diseases were not considered). In addition, other genes that were not included among the canonical pathway enriched-genes were also mentioned in the main text by considering their potential relevance in context. Additional metabolic pathway analyses were performed using the BioCyc database (https://biocyc.org, Menlo Park, CA, USA). The heatmaps were generated by GraphPad Prism software Version 7 for Windows (GraphPad Software, San Diego, CA, USA).

## Figures and Tables

**Figure 1 ijms-22-01330-f001:**
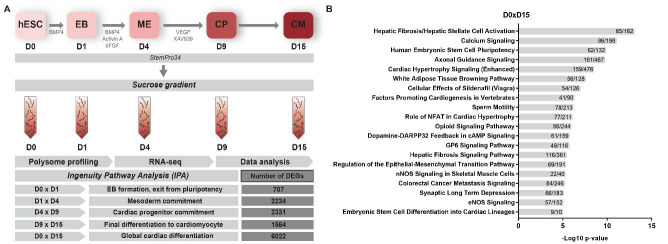
In vitro cardiac differentiation and data analysis. (**A**) The experimental procedure consisted of human embryonic stem cell (hESC) cardiac differentiation, polysome sample collection for RNA-seq and data analysis. hESCs, human embryonic stem cells. EB, embryoid body. ME, cardiac mesoderm. CP, cardiac progenitors. CM, cardiomyocyte. (**B**) Top 20 IPA canonical pathways enriched in the D0 × D15 transition. The ratios between the number of differentially expressed genes (DEGs) found and the total number of genes of each canonical pathway are shown inside the bars.

**Figure 2 ijms-22-01330-f002:**
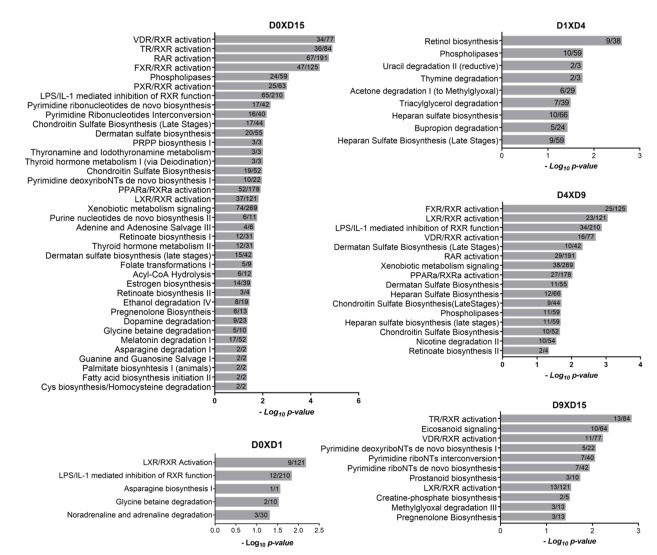
Metabolism-related canonical pathways enriched in in vitro cardiac differentiation. Enriched Ingenuity Pathway Analysis (IPA) canonical pathways related to metabolism at each in vitro cardiac differentiation transition. The ratios between the number of DEGs found and the total number of genes of each canonical pathway are shown inside the bars.

**Figure 3 ijms-22-01330-f003:**
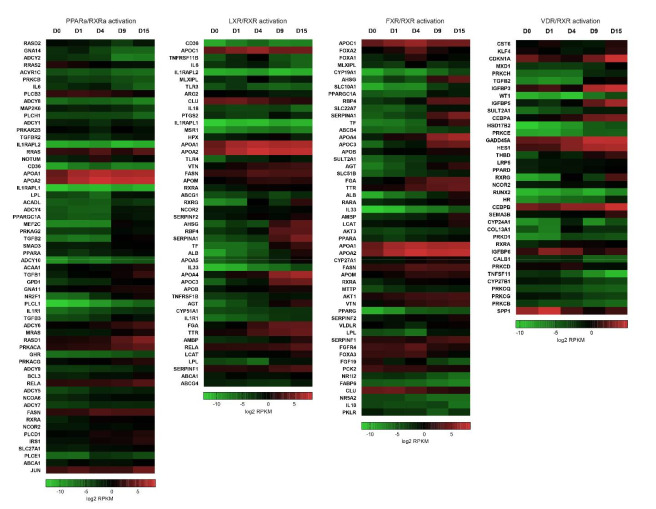
Retinoic X receptor (RXR) heterodimer pathways enriched in in vitro cardiac differentiation. Heatmaps representing the gene expression profiles (log2 RPKM values) of DEGs related to the PPAR/RXR, LXR/RXR, FXR/RXR, and VDR/RXR activation pathways.

**Figure 4 ijms-22-01330-f004:**
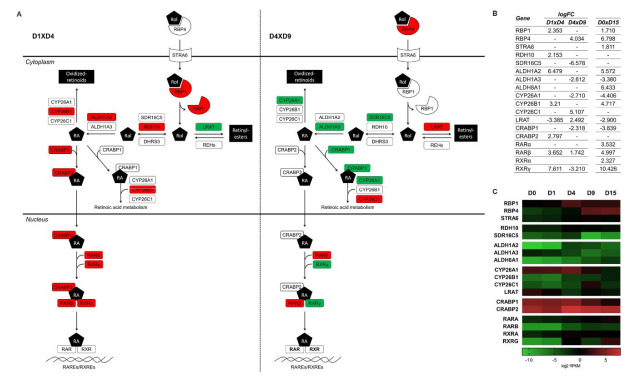
Retinoic acid (RA) synthesis and signaling activation during in vitro cardiac differentiation. (**A**) Comparison of DEGs identified in the RA signaling pathway at the mesoderm (D1 × D4) and cardiac progenitor (D4 × D9) commitment stages. (**B**) Fold change values (log) of the DEGs from the D1 × D4, D4 × D9 and D0 × D15 comparisons. (**C**) Heatmaps representing the gene expression profiles (log2 RPKM values) of DEGs related to RA signaling. Red boxes indicate upregulated genes, while green boxes indicate downregulated genes. Rol: retinol, Ral: retinal. Adapted from Ingenuity Pathway Analysis.

**Figure 5 ijms-22-01330-f005:**
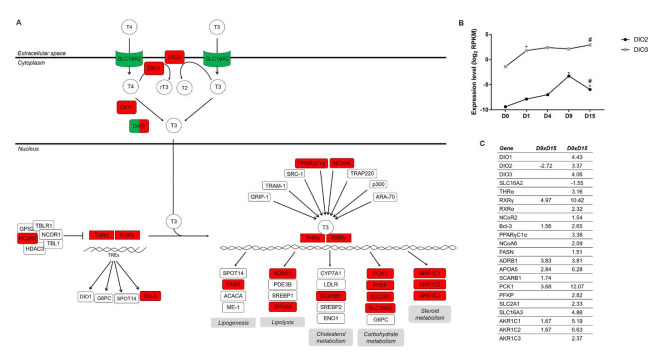
Regulation of TR/RXR activation in in vitro cardiac differentiation. (**A**) Scheme of the TR/RXR activation pathway highlighting the DEGs identified at the final cardiomyocyte commitment (D9 × D15) and the D0 × D15 stages for comparison. (**B**) Expression levels (log2 RPKM) of DIO2 and DIO3. Comparison with the ^*^ preceding time point or ^#^ D0 according to −1.5 ≥ logFC ≥ 1.5 (FDR ≤ 0.05). (**C**) Fold change values (log) of genes identified in the TR/RXR pathway in the D9 × D15 and D0 × D15 comparisons. Red boxes indicate upregulated genes, while green boxes indicate downregulated genes. Red+green boxes indicate that the gene is upregulated and downregulated at different time points. Adapted from Ingenuity Pathway Analysis.

**Figure 6 ijms-22-01330-f006:**
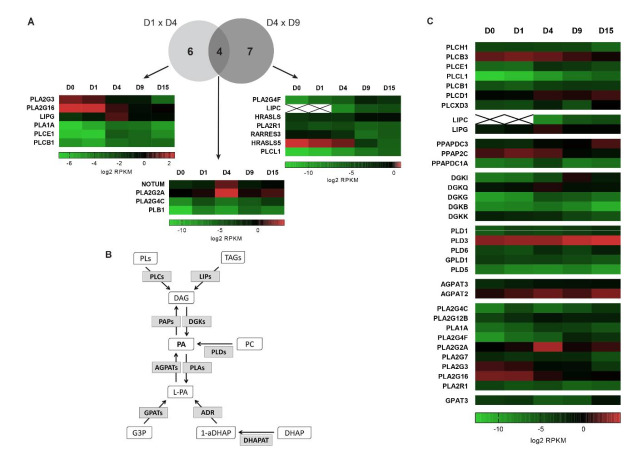
Lipid metabolism-related genes regulated in in vitro cardiac differentiation. (**A**) Comparison of genes identified in the “phospholipase” canonical pathway at D1 × D4 and D4 × D9. Venn diagram showing the number of DEGs. (**B**) Scheme of phosphatidic acid (PA) metabolism (Adapted from [84]). (**C**) Heatmaps representing the gene expression profiles (log2 RPKM values) of DEGs related to PA metabolism.

**Figure 7 ijms-22-01330-f007:**
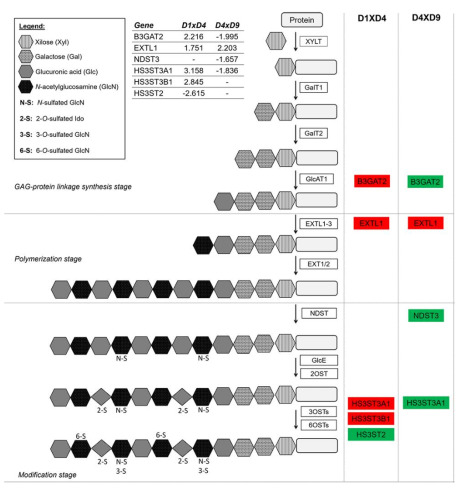
Differentially expressed genes involved in heparan sulfate biosynthesis. Comparison of DEGs identified in the heparan sulfate biosynthesis pathway at the mesoderm (D1 × D4) and cardiac progenitor (D4 × D9) commitment stages. Red boxes indicate upregulated genes, while green boxes indicate downregulated genes. XYLT: xylosyltransferase, GalT: galactosyltransferase, GlcAT1: glucuronic acid transferase, NDST: *N*-deacetylase/*N*-sulfotransferase-1, GlcE: glucuronic acid epimerase, 2OST: 2-O-sulfotransferase, 3OST: 2-O-sulfotransferase, 6OST: 2-O-sulfotransferase (adapted from [98]).

**Figure 8 ijms-22-01330-f008:**
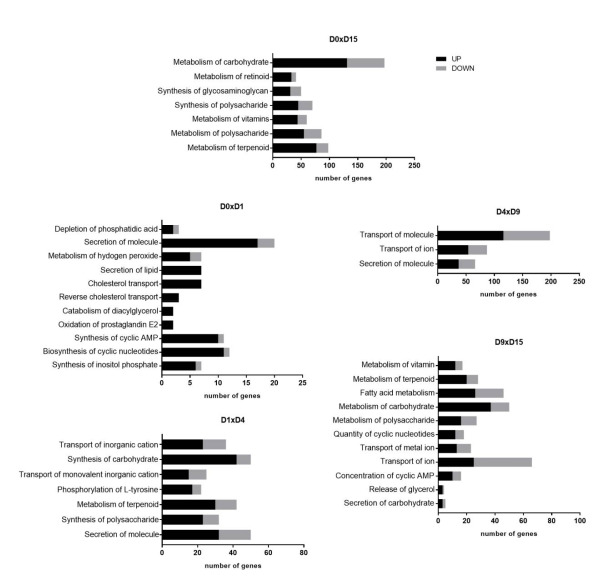
Biological functions related to metabolism enriched in in vitro cardiac differentiation. Graphs show the number of DEGs (up- and downregulated) identified based on the terms obtained from IPA at each cardiac differentiation transition.

**Figure 9 ijms-22-01330-f009:**
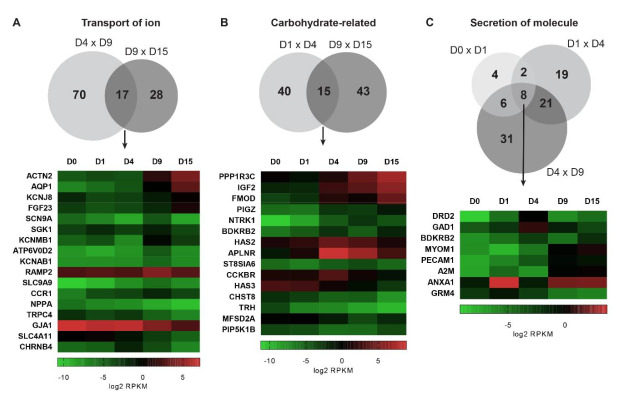
Biological processes that are highly regulated during in vitro cardiac differentiation. Comparison of DEGs enriched for terms related to “Transport of ion”, “Carbohydrate” and “Secretion of molecules” at different time points. Venn diagrams show the number of DEGs, and the heatmaps represent the gene expression profiles (log2 RPKM values) of common DEGs identified in each comparison.

## Data Availability

The data presented in this study are openly available in Short Read Archive, NCBI at https://www.ncbi.nlm.nih.gov/sra/SRP150416.

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
