# Peer review of "Reorganization of Metabolism during Cardiomyogenesis Implies Time-Specific Signaling Pathway Regulation"

_ijms, 2021, doi:10.3390/ijms22031330_

Round 1

Reviewer 1 Report

In the proposed manuscript Barison and colleagues analyzed the metabolic expression profile during cardiac differentiation of human embryonic stem cells (hESCs) by analyzing the polysome-bound mRNA-seq. Over the 15 day differentiation protocol, the authors confirm the immaturity of hESC-cardiomyocyte energy metabolism. Among the entire expression profile, the authors highlighted the retinoic acid, lipid homeostasis and extracellular matrix-related pathways. The study offers a detailed metabolic profile of hESC-cardiomyocytes over 15 day differentiation. I have the following suggestions:

  • I would suggest to extrapolate a panel from the gene expression profile, reporting time course of structural/sarcomeric genes, ion channel-genes and calcium handling genes overtime, to show the progression of cardiac differentiation, although after 15 days of differentiation in 2D CMs are still immature as expected.
  • The authors should structure the discussion by considering and emphasizing the media composition used to culture and differentiate the cells, as it strongly affects and modulates cardiomyocyte metabolism.
  • How do the authors place their data in light of the recent published paper by Feyen D.A.M et al., 2020 ? (Feyen DAM, McKeithan WL, Bruyneel AAN, et al. Metabolic Maturation Media Improve Physiological Function of Human iPSC-Derived Cardiomyocytes. Cell Rep. 2020;32(3):107925. doi:10.1016/j.celrep.2020.107925). This is a relevant paper that I suggest the authors to include in the discussion as it shows how metabolic modulation results in in vitro CM maturity.

Author Response

In the proposed manuscript Barison and colleagues analyzed the metabolic expression profile during cardiac differentiation of human embryonic stem cells (hESCs) by analyzing the polysome-bound mRNA-seq. Over the 15 day differentiation protocol, the authors confirm the immaturity of hESC-cardiomyocyte energy metabolism. Among the entire expression profile, the authors highlighted the retinoic acid, lipid homeostasis and extracellular matrix-related pathways. The study offers a detailed metabolic profile of hESC-cardiomyocytes over 15 day differentiation. I have the following suggestions:

  • I would suggest to extrapolate a panel from the gene expression profile, reporting time course of structural/sarcomeric genes, ion channel-genes and calcium handling genes overtime, to show the progression of cardiac differentiation, although after 15 days of differentiation in 2D CMs are still immature as expected.

As suggested by the reviewer, we included a supplementary figure (Figure S1) showing a heatmap of the expression levels of some cardiac markers, such as cardiac regulators (GATA4, MEF2C, TBX5, ISL1, NKX2-5, NR2F2, IRX4, VCAM1), structural/sarcomeric genes (TNNC1, TNNI1, TNNI2, TNNI3, TNNT2, MYH6, MYH7, MYL3, MYL4, MYL7), calcium handling genes (SERCA2, CACNA1D, CACNA1G) and ion-channel genes (KCNA5, KCNJ3, KCNJ5, KCNK3, KCNQ1). As expected, the expression pattern of those genes is in agreement with the progression of cardiac differentiation, showing higher levels at D9 and D15.

  • The authors should structure the discussion by considering and emphasizing the media composition used to culture and differentiate the cells, as it strongly affects and modulates cardiomyocyte metabolism.

In accordance with the reviewer observation, we discuss how culture medium influences the cardiomyocyte metabolism during the section: “2.1.2 Energy metabolism changes between hESCs and hESCs-CM”. Specifically, in lines 161-166 the effect of high glucose concentration (approx. 25 mM) on differentiation media (StemPro-34) is discussed, focusing mainly on how the high glucose availability impairs the metabolic switch, producing more immature cardiomyocytes. Also, attending to the reviewer suggestion, we include this topic in the corrected version of the manuscript, in the section “2.3 Additional lipid metabolism-related pathways are enriched during cardiac differentiation”, lines 562-566: “Again, differentiation medium composition seems to be influencing cardiomyocyte metabolism. As previously described, StemPro-34 presents high glucose concentrations (about 25 mM = 4.5 g/L), while low lipid contribution (around 5 mg/L). In this way, glycolytic intermediates as G3P could be used by CMs to direct the synthesis of PA, a precursor for lipid biosynthesis”.

  • How do the authors place their data in light of the recent published paper by Feyen D.A.M et al., 2020 ? (Feyen DAM, McKeithan WL, Bruyneel AAN, et al. Metabolic Maturation Media Improve Physiological Function of Human iPSC-Derived Cardiomyocytes. Cell Rep. 2020;32(3):107925. doi:10.1016/j.celrep.2020.107925). This is a relevant paper that I suggest the authors to include in the discussion as it shows how metabolic modulation results in in vitro CM maturity.

The recent paper cited by the reviewer presents very interesting data regarding the influence on culture media on iPSC-derived cardiomyocytes. Feyen and colleagues propose a maturation media for cardiomyocytes that increases fatty-acid oxidation and also enhances electrophysiological and mechanical parameters leading to a more mature phenotype. We recognize that cardiomyocyte maturation is a decisive aspect to model cardiac diseases, and that metabolic modulation can contribute to that, although this was not our focus in this work. The objective of our paper is to understand the metabolic remodeling during differentiation, focusing on early and intermediate steps from pluripotent cells to cardiomyocyte commitment. Nevertheless, and as suggested by the reviewer, this topic was included in our discussion in lines 166-168: “Also, Feyen and collaborators demonstrated that the inclusion of oxidative substrates to the culture medium of iPSCs-CM produce metabolically mature CMs, improving electrophysiological and mechanical parameters (Feyen et al. 2020)”.

References

Cao, Feng, Roger A Wagner, Kitchener D Wilson, Xiaoyan Xie, Ji-dong Fu, Micha Drukker, Andrew Lee, et al. 2008. “Transcriptional and Functional Profiling of Human Embryonic Stem Cell-Derived Cardiomyocytes.” PLoS ONE 3 (10): e3474. https://doi.org/10.1371/journal.pone.0003474.

Chung, Susan, Petras P. Dzeja, Randolph S. Faustino, Carmen Perez-Terzic, Atta Behfar, and Andre Terzic. 2007. “Mitochondrial Oxidative Metabolism Is Required for the Cardiac Differentiation of Stem Cells.” Nature Clinical Practice Cardiovascular Medicine 4 (SUPPL. 1): 1–12. https://doi.org/10.1038/ncpcardio0766.

Feyen, Dries A.M., Wesley L. McKeithan, Arne A.N. Bruyneel, Sean Spiering, Larissa Hörmann, Bärbel Ulmer, Hui Zhang, et al. 2020. “Metabolic Maturation Media Improve Physiological Function of Human IPSC-Derived Cardiomyocytes.” Cell Reports 32 (3): 107925. https://doi.org/10.1016/j.celrep.2020.107925.

Lian, Xiaojun, Jianhua Zhang, Samira M. Azarin, Kexian Zhu, Laurie B. Hazeltine, Xiaoping Bao, Cheston Hsiao, Timothy J. Kamp, and Sean P. Palecek. 2013. “Directed Cardiomyocyte Differentiation from Human Pluripotent Stem Cells by Modulating Wnt/β-Catenin Signaling under Fully Defined Conditions.” Nature Protocols 8 (1): 162–75. https://doi.org/10.1038/nprot.2012.150.

Tohyama, Shugo, Fumiyuki Hattori, Motoaki Sano, Takako Hishiki, Yoshiko Nagahata, Tomomi Matsuura, Hisayuki Hashimoto, et al. 2013. “Distinct Metabolic Flow Enables Large-Scale Purification of Mouse and Human Pluripotent Stem Cell-Derived Cardiomyocytes.” Cell Stem Cell 12 (1): 127–37. https://doi.org/10.1016/j.stem.2012.09.013.

Reviewer 2 Report

The story is interesting and sheds new light on the reorganization of metabolism during cardiomyogenesis, but the presented work lacks important data to make the conclusion here.

Major points are:

·       What about later time points of cardiac differentiation (d15 is pretty early): Please include d30 and d60 in your experiments.

·       The authors always write sentences like these:  the gene expression profile confirmed that hESC-CMs differentiated for 15 days show immature energy metabolism that is highly dependent on glycolysis. That is already known and not new at day 15 of differentiation and was shown before several times.

·       Can the authors confirm the sequencing results on mRNA or protein level of stage-associated metabolism markers (qPCR/Western Blot)? Please confirm sequencing data on mRNA or protein.

·       The whole work is descriptive. What can the reader get from Figures 2, 3, 8, 9 e.g.? The authors should more focus on specific ways and single genes/proteins and confirm the sequencing data by molecular analysis.

·       Figures are too small, especially Figure 1 and 2, 8.

·       Include numbers of DEEs in the figures.

Author Response

The story is interesting and sheds new light on the reorganization of metabolism during cardiomyogenesis, but the presented work lacks important data to make the conclusion here.

Major points are:

  • What about later time points of cardiac differentiation (d15 is pretty early): Please include d30 and d60 in your experiments.

We thank the reviewer for the suggestion and agree that it would be very interesting to follow the metabolism changes through later differentiation stages. However, our main goal in this work was to investigate early and intermediate time points during the cardiomyogenic commitment. The majority of already published works in the field focused on the study and comparison of ESC-derived cardiomyocytes with undifferentiated cells or fetal hearts, which are very distinct and developmentally-distant cells. None of them investigated metabolic pathway changes between early and intermediate states. It’s well established that in vitro derived cardiomyocytes are metabolic and functional immature, and that there are considerable changes during long time in culture. However, it has not yet been accessed whether the signaling for the energy metabolism switch and other pathways dependent on and/or influenced by general cellular metabolism could begin in early cardiac commitment. For this reason, we chose not to include later time points in the analysis.

  • The authors always write sentences like these:  the gene expression profile confirmed that hESC-CMs differentiated for 15 days show immature energy metabolism that is highly dependent on glycolysis. That is already known and not new at day 15 of differentiation and was shown before several times.

We agree with the reviewer that the fact that hESC-CMs are metabolically immature is widely described in the literature. Indeed, our discussion about energy metabolism is based in several papers describing this topic (Chung et al. 2007; Tohyama et al. 2013; Cao et al. 2008). We decided to include the results related to energy metabolism because it is a relevant data when the discussion is about cell metabolism, and in this case, during cardiac differentiation. Additionally, the fact that these data are already widely known, was an approach to corroborate our experimental and RNA-sequencing data. Having confirmed that hESC-CMs present the metabolic profile expected to our experimental conditions, we focused on exploring specific metabolic and signaling pathways regulating both earlier and intermediate stages of cardiomyogenesis, not only the end point of differentiation, the hESCs-CMs.        

  • Can the authors confirm the sequencing results on mRNA or protein level of stage-associated metabolism markers (qPCR/Western Blot)? Please confirm sequencing data on mRNA or protein.

As suggested by the reviewer, sequencing data was confirmed by qPCR. We focused on the main pathways discussed in our work: retinoic acid metabolism, thyroid hormone metabolism and heparan sulfate biosynthesis. Some differentially expressed genes from each pathway and at specific stages of cardiomyogenesis were validated by its relative expression to the normalizer gene GAPDH. The results are represented along the corresponding expression level (RPKM) obtained by RNA-seq experiments. As expected, the expression pattern obtained by qPCR overlaps with the presented in this work by sequencing data (Figures S5A, S6 and S7). Additionally, the expression of the highly regulated gene RXRG was further validated using a second model of cardiac differentiation: the monolayer protocol (Lian et al. 2013). Comparing its expression between D0 and D15 by qPCR we verified the same expression pattern than the obtained by RNA-seq: higher expression at cardiomyocytes stage (Figure S5B). This result was also validated evaluating the protein expression level of RXRγ between D0 and D15 with the monolayer protocol, by immunofluorescence assay (Figure S5C).   

  • The whole work is descriptive. What can the reader get from Figures 2, 3, 8, 9 e.g.? The authors should more focus on specific ways and single genes/proteins and confirm the sequencing data by molecular analysis.

One of the main goals of this work was to deeper explore the polysome-associated RNA-seq data of human cardiomyogenesis. As an inherent aspect of high throughput data, valuable information can be extracted from tons of crude data. We agree that the work is essentially descriptive, but it could be the first step from uncovering a crucial cellular process, e.g., since it represents the screening for those valuable informations. Between hundreds of significantly enriched Canonical Pathways terms from IPA (e.g. 178 from D0 vs D15 analysis), we were able to point out and deeply discuss five groups of genes or signaling pathways regulated during the cardiac differentiation and related to metabolism. The current literature in the field has been focusing on the differences of ESC-derived cardiomyocytes and undifferentiated cells or fetal hearts, mainly regarding the energetic metabolism. We chose to focus on earlier and intermediate stages of differentiation and highlight the time-specific signaling of metabolism-related pathways, beyond and not exclusively related to energetic metabolism. In Fig 2, we show the metabolism-related Canonical Pathways enriched in our data, which gives the reader an idea of the variety of pathways regulated during differentiation and also enables the identification of pathways regulated in more than one or that are exclusive from a specific time-point comparison. In Fig 3, we show heatmaps of DEGs from RXR-related signaling pathways which give the reader the information about expression patterns of each DEG during the differentiation, enable the identification of common genes between different pathways, and allow the reader to find a specific gene of interest, for example. In Fig 8, we show the Biological Function terms related to metabolism (and the number of genes) found as enriched in each time-point comparison, which give the reader the information of more predominant regulated functions related to carbohydrates compared to lipids or transport of molecules, e.g.,  in a specific time-point comparison or even between time-point comparisons. Finally, Fig 9 complements Fig 8 analysis and shows some examples of genes highly regulated during differentiation that are involved in important metabolism-related biological functions.

As suggested by the reviewer and described in the previous answer, sequencing data was validated by qPCR and immunofluorescence assays, corroborating the expression patterns described along our work.

 To the best of our knowledge, there is not any work that gives an overview of metabolism-related pathways regulation in distinct developmental stages of human cardiomyogenesis and discusses detailed and valuable information of metabolites and metabolic processes directly associated with cellular differentiation. This work represents a solid ground for further functional and more specific studies that may elucidate crucial and specific regulatory processes. We believe that, besides essentially descriptive, our work is exceedingly important for the field.

  • Figures are too small, especially Figure 1 and 2, 8.

Attending the reviewer observation, the figures have been revised to make them more visible to the reader. Specially, figures 1, 2 and 8.

  • Include numbers of DEEs in the figures

As suggested by the reviewer, we included the total number of DEGs found at each transition of cardiomyogenic differentiation in Figure 1A. Also, we included the ratio between the number of DEGs and the total of genes in each pathway in Figure 1B, which shows the pathways enriched at D0xD15; and in Figure 2, where the canonical pathways related to metabolism are shown. For additional information about the number, logFC and p-values of DEGs found in the pathways discussed along the manuscript (RA and TH metabolism, lipid-related pathways and HS biosynthesis), please see supplementary tables S2 and S5-8. 

References

Cao, Feng, Roger A Wagner, Kitchener D Wilson, Xiaoyan Xie, Ji-dong Fu, Micha Drukker, Andrew Lee, et al. 2008. “Transcriptional and Functional Profiling of Human Embryonic Stem Cell-Derived Cardiomyocytes.” PLoS ONE 3 (10): e3474. https://doi.org/10.1371/journal.pone.0003474.

Chung, Susan, Petras P. Dzeja, Randolph S. Faustino, Carmen Perez-Terzic, Atta Behfar, and Andre Terzic. 2007. “Mitochondrial Oxidative Metabolism Is Required for the Cardiac Differentiation of Stem Cells.” Nature Clinical Practice Cardiovascular Medicine 4 (SUPPL. 1): 1–12. https://doi.org/10.1038/ncpcardio0766.

Feyen, Dries A.M., Wesley L. McKeithan, Arne A.N. Bruyneel, Sean Spiering, Larissa Hörmann, Bärbel Ulmer, Hui Zhang, et al. 2020. “Metabolic Maturation Media Improve Physiological Function of Human IPSC-Derived Cardiomyocytes.” Cell Reports 32 (3): 107925. https://doi.org/10.1016/j.celrep.2020.107925.

Lian, Xiaojun, Jianhua Zhang, Samira M. Azarin, Kexian Zhu, Laurie B. Hazeltine, Xiaoping Bao, Cheston Hsiao, Timothy J. Kamp, and Sean P. Palecek. 2013. “Directed Cardiomyocyte Differentiation from Human Pluripotent Stem Cells by Modulating Wnt/β-Catenin Signaling under Fully Defined Conditions.” Nature Protocols 8 (1): 162–75. https://doi.org/10.1038/nprot.2012.150.

Tohyama, Shugo, Fumiyuki Hattori, Motoaki Sano, Takako Hishiki, Yoshiko Nagahata, Tomomi Matsuura, Hisayuki Hashimoto, et al. 2013. “Distinct Metabolic Flow Enables Large-Scale Purification of Mouse and Human Pluripotent Stem Cell-Derived Cardiomyocytes.” Cell Stem Cell 12 (1): 127–37. https://doi.org/10.1016/j.stem.2012.09.013.

Round 2

Reviewer 1 Report

The revised manuscript has been significantly improved and now warrants publication in IJMS.

Author Response

The reviewer 1 has approved the article for publication.  

Reviewer 2 Report

The data shown in the revised version are of high interest. The authors have nicely clarified and presented some concerns of the 1st review.

But there are still some minor points, which have to be addressed:

-The authors have included the validation of sequencing data by qPCR and immunofluorescence stainings in figures S5, S6 and S7 and stated in the qPCR mehods that each gene was assayed in technical and biological triplicates. What means biological triplicates in this case? Three different differentiations? Plesae clarify. Here, it would be better to show dots in the bars.

-Figure S5A, S6, S7: What means ‘fold to GapDH’? How have the authors calculated the expression? Have the authors make a fold change to GAPDH or used 2^ (-delta ct)? Please clarify.

-DEGs were included in figures 1A, but not differentially expressed exons (DEEs) as suggested in the 1st review. Are DEEs available? If yes, it would be nicely to include these data, because of highly new impact.

-Figure 1 was enlarged a bit, Figures 2 and 8 are the same as before.

Author Response

Response to reviewer 2 (Round 2)

The data shown in the revised version are of high interest. The authors have nicely clarified and presented some concerns of the 1st review.

But there are still some minor points, which have to be addressed:

  • The authors have included the validation of sequencing data by qPCR and immunofluorescence stainings in figures S5, S6 and S7 and stated in the qPCR mehods that each gene was assayed in technical and biological triplicates. What means biological triplicates in this case? Three different differentiations? Please clarify. Here, it would be better to show dots in the bars.

Yes, biological triplicates mean three independent differentiations. We clarified this issue in the “Supplementary Material and Methods”, at the “Quantitative real time PCR” topic. As suggested by the reviewer, we included dots in the bars of figures S5, S6 and S7.

  • Figure S5A, S6, S7: What means ‘fold to GapDH’? How have the authors calculated the expression? Have the authors make a fold change to GAPDH or used 2^ (-delta ct)? Please clarify.

qPCR results shown as “fold to GAPDH” were calculated using the formula 2^(-delta Ct) and means how much that gene is expressed when compared to GAPDH expression. This way of analysis is distinct of using the formula 2^ (-delta delta Ct), which stablishes a time-point/condition for relative expression. We add this information in the “Supplementary Material and Methods”, at the “Quantitative real time PCR” topic.

  • DEGs were included in figures 1A, but not differentially expressed exons (DEEs) as suggested in the 1st Are DEEs available? If yes, it would be nicely to include these data, because of highly new impact.

Unfortunately, that information (differentially expressed exons) is not available.

  • Figure 1 was enlarged a bit, Figures 2 and 8 are the same as before.

The font sizes in both figures (2 and 8) were increased to better visualization.